# Sorption of Cd²⁺ on Bone Chars with or without Hydrogen Peroxide Treatment under Various Pyrolysis Temperatures: Comparison of Mechanisms and Performance

**Qing Guo** [1,2]**, Hongmei Tang** [1,2]**, Lu Jiang** [1,2]**, Meiqing Chen** [1,2]**, Nengwu Zhu** [1,2] **and Pingxiao Wu** [1,2,3,]*****

1    School of Environment and Energy, South China University of Technology, Guangzhou 510006, China;
     guoqing1723@163.com (Q.G.); ttthongmei@163.com (H.T.); liangju1634@163.com (L.J.);
     cmq2021@scut.edu.cn (M.C.); nwzhu@scut.edu.cn (N.Z.)
2    The Key Lab of Pollution Control and Ecosystem Restoration in Industry Clusters, Ministry of Education,
     Guangzhou 510006, China
3    Guangdong Provincial Key Laboratory of Solid Wastes Pollution Control and Recycling,
     Guangzhou 510006, China
*    Correspondence: pppxwu@scut.edu.cn

**Abstract:** In this study, bone char pretreated with hydrogen peroxide and traditional pyrolysis was applied to remove Cd²⁺ from aqueous solutions. After hydrogen peroxide pretreatment, the organic matter content of the bone char significantly decreased, while the surface area, the negative charge and the number of oxygen-containing functional groups on the bone char surface increased. After being pyrolyzed, the specific surface area and the negative charge of the material were further improved. The adsorption kinetics and isotherms of Cd²⁺ adsorption were studied, and the influence of solution pH and the presence of ionic species were investigated. The experimental results showed that the samples with lower crystallinity exhibited less organic matter content and more surface oxygen-containing functional groups, resulting in stronger adsorption capacity. After being treated with hydrogen peroxide and pyrolyzed at 300 °C, the maximum adsorption capacity of bone char was 228.73 mg/g. The bone char sample with the lowest adsorption capacity(47.71 mg/g) was pyrolyzed at 900 °C without hydrogen peroxide pretreatment. Ion exchange, surface complexation, and electrostatic interactions were responsible for the elimination of Cd²⁺ by the bone char samples. Overall, this work indicates that hydrogen peroxide-treated pyrolytic bone char is a promising material for the immobilization of Cd²⁺.

**Keywords:** bone char; hydrogen peroxide treatment; cadmium; adsorption mechanism; ion interchange

## 1. Introduction

In recent years, with the development of industries such as metallurgy, printing, mining, etc., the pollution of toxic trace metal has attracted much attention. Cadmium (Cd) has a long-term effect on aquatic environments [1], plants, and human health due to its high mobility and the difficulty associated with its removal [2,3]. Cd²⁺ can enter organisms through the respiratory tract and other organs, and it aggregates in the liver and other organs, and eventually has a significantly negative impact on the metabolism and growth of organisms [4,5]. Therefore, many researchers have proposed various methods for the removal of Cd²⁺, including chemical precipitation and adsorption [6–9]. Among these proposed methods, adsorption is a widely utilized technology due to its easy operation and low cost. Carbon materials [10], nano-ferromagnetic materials [11], iron manganese oxides [12], clay minerals [13], biomass materials [14], modified biomaterials [15], and composite materials [16] are often employed to treat effluent.

Bone char (BC) is a common adsorbent, which is manufactured from waste biomaterial. Compared to other adsorbents, bone char has adequate availability, low cost, and high

specific adsorption capacity toward different pollutants, including fluoride [17], arsenates [9], and heavy metals [18–21] such as $Co^{2+}$, $Cu^{2+}$, $Cd^{2+}$, $Ni^{2+}$, and $Zn^{2+}$. For example, the adsorption capacity of bone-synthesized BC to $Cd^{2+}$ was generally 130.8–213.0 mg/g. However, the physicochemical properties of pristine bone char demonstrate significant heterogeneity. According to the feedstock and treatment of bone char, the materials exhibit widely varying specific surface areas, porosity, and surface charge [22]. Furthermore, the mechanisms that control the mobility of $Cd^{2+}$ and other pollutants by bone char are significantly affected by material properties [23]. Therefore, optimizing the material properties of bone char plays a key role in enhancing its adsorption capacity.

It is recognized that pyrolysis can improve the physicochemical properties of bone char, thereby improving its sorption capacity of $Cd^{2+}$ [24]. The pyrolysis temperature is the key to controlling the properties of bone char. Previous studies showed that most unstable organic components were degraded and porosity was improved under low-temperature pyrolysis. However, high-temperature pyrolysis promoted the crystallization of hydroxyapatite into large particles, thus reducing the porosity of BC [21,25]. In addition, the major problem with the method was non-uniform during the process of pyrolysis, which resulted from internal organic matter not being degraded completely. Several reports have shown that hydrogen peroxide removes internal organic material from bone to create a porous structure that increases specific surface area [26,27]. Meanwhile organic matter could be converted into oxygen-containing functional groups, the density of which could correspondingly improve the adsorption capacity of heavy metal [28]. Therefore, being pretreated with hydrogen peroxide and pyrolysis is an efficient method for BC to enhance its adsorption of heavy metals.

In this work, fishbone powder was pretreated with $H_2O_2$ and pyrolyzed at different temperatures to produce the adsorbent for the elimination of $Cd^{2+}$ from an aqueous solution. The purpose of this study was as follows: (1) to study the effects of $H_2O_2$ and pyrolysis temperature on bone char characteristics; (2) to investigate the adsorption of $Cd^{2+}$ in the presence of other environmental species; (3) to investigate the $Cd^{2+}$ adsorption mechanisms of the bone char materials; (4) to establish a comprehensive relationship between bone char treatment, adsorption capacity, and adsorption mechanism.

## 2. Materials and Methods

### 2.1. Materials and Reagents

The bones, obtained from the backbone of a fish, were bought from a Panyu seafood market (Guangzhou, Guangdong Province, China). All chemicals were of analytical grade and were used without further purification. Nitric acid ($HNO_3$) and sodium hydroxide (NaOH) were bought from Aladdin Chemistry Co. Ltd. (Shanghai, China). Sodium fluoride (NaF) and sodium chloride (NaCl) were provided by Sinopharm Chemical Reagent Co. Ltd. (Shanghai, China).

### 2.2. Preparation of Bone Char with/without $H_2O_2$ Pretreatment

The raw fishbone was dried at 80 °C for 24 h. A portion of the material was ground through a 200 mesh screen and called RB. The other portion of the material was pyrolyzed in a tube furnace in a nitrogen atmosphere at different temperatures (300 °C, 500 °C, 700 °C, and 900 °C) for 2 h without $H_2O_2$ pretreatment. These obtained bone char materials were denoted BC-300, BC-500, BC-700, and BC-900 based on their pyrolysis temperatures. To investigate the effect of $H_2O_2$ pretreatment, the dried bone powder was treated with $H_2O_2$ (30%), then was pyrolyzed in a tube furnace in a nitrogen atmosphere under 300 °C for 2 h. The obtained bone char material was denoted BCH-300 based on its pyrolysis temperature. An unpyrolyzed bone specimen that was pretreated with $H_2O_2$ was denoted BCH. After pyrolysis, the samples were washed with DI water. The washed materials were then dried at 80 °C for 12 h. Finally, the dried materials were passed through a 200-mesh sieve and were stored in a sealed container.

### 2.3. Characterization

Various characterization techniques were used to measure the physical and chemical characteristics of the adsorbents. Details were shown in the Supplementary Materials.

### 2.4. Batch Experiments and Analytical Methods

Detailed information on the adsorption tests and analytical methods were described in the Supplementary Materials.

### 2.5. Desorption Study

Leach solutions of pH 1, 2, 4, and 6 were prepared 0.1 mol/L $HNO_3$ and distilled in water. Besides, 25 and 100 mmol/L $Ca(NO_3)_2$ solutions were prepared.

## 3. Results and Discussion

### 3.1. Characterization of BC and BCH Samples

The morphological characterization and the size of the BC and BCH samples were shown in Figure 1. The surface morphology of BC-300 only exhibits a few cracks, while the BCH-300 sample has a rough and scattered appearance. This suggests that $H_2O_2$ treatment fragments the granular pristine bone chars into ultrafine particles. This occurs due to the removal of organic constituents by $H_2O_2$. The images obtained at a magnification level of 100,000× clearly show the morphologies of the samples in detail. A typical neat spherical porous structure appears with increasing pyrolysis temperature. Moreover, increasing the pyrolysis temperature enhances the formation of hydroxyapatite crystals, particularly for the highest pyrolysis temperature. The morphology of BC-900 is almost the same as highly crystalline hydroxyapatite [9]. Higher pyrolysis temperatures may lead to higher crystallinity in these samples.

The specific surface area (SSA) values of the BCH samples are larger than those of the BC samples (Table S1). The BC samples have SSA values ranging from 50.24–158.47 $m^2/g$, while the SSA values of the BCH samples rise to 172.66 $m^2/g$ due to the $H_2O_2$ pretreatment [21,29]. The SSA values of the BC samples increase with increasing pyrolysis temperature from 300 to 700 °C. This is due to the existence of exposed pores produced by the discharge of volatiles during pyrolysis of the non-pretreated BC samples. The SSA of BC-700 is 158.478 $m^2/g$, while the SSA of BC-900 is 108.24 $m^2/g$. Thus, an excessively high pyrolysis temperature reduces the specific surface area of the bone char, which is due to material contraction and crystallization [21]. In addition, pyrolysis can further improve the SSA value of BCH samples, which may be due to the further decrease in material organic matter. These results were confirmed by dissolved organic carbon (DOC) and Energy Dispersive Spectroscopy (EDS) (Table S1) [30].

The Thermo Gravimetric Analyzer (TGA) results (Figure S2) indicate that the mass of the BC samples significantly decreases between 200 and 500 °C, indicating that large amounts of organic matter are pyrolyzed in this temperature range [22]. The DOC results also demonstrate that the organic matter content of the BC samples significantly decreases with increasing pyrolysis temperature [17], while the level of dissolved organic carbon in BCH samples are very low. As shown in Table S1, the concentrations of Ca (wt%), P (wt%), and O (wt%) in the samples show significant variation, which may be due to the decline in the concentration of C (wt%). Furthermore, the rise in the concentration of O (wt%) might be attributed to the transformation of –$CH_3$ structures to oxygen-containing groups. Moreover, BCH-300 also shows a significant decrease in average pore diameter compared with RB, which may be attributed to pyrolysis removing the internal organic matter. This trend is also visible in the SEM images shown in Figure 1.

FT-IR analysis was performed to investigate the surface functional groups of the BC and BCH samples. As displayed in Figure 2a, the stretching vibration centered at 3400 $cm^{-1}$ is the characteristic peak of –OH. OH-containing groups are regarded as the typical function groups of biochar and hydroxyapatite [31]. The bands at 1425 and 1633 $cm^{-1}$ are likely caused by the tensile oscillation of –COO or $CO_3^{2-}$ of species adsorbed from the air [31,32].

The FT-IR peaks in the range of 450 to 1100 cm$^{-1}$ can be ascribed to $PO_4^{3-}$ groups in all the BC and BCH materials [33]. The adsorption band at 1040 cm$^{-1}$ is attributed to P–O tensile oscillation. The bands at 964 cm$^{-1}$ and 606 cm$^{-1}$ correspond to O–P–O bending oscillations and O–P–O bending vibrations [32], respectively. The band at 565 cm$^{-1}$ is caused by the calcium in the inorganic structure of the bone char. The presence of this band indicates the linkage between calcium and phosphate groups.

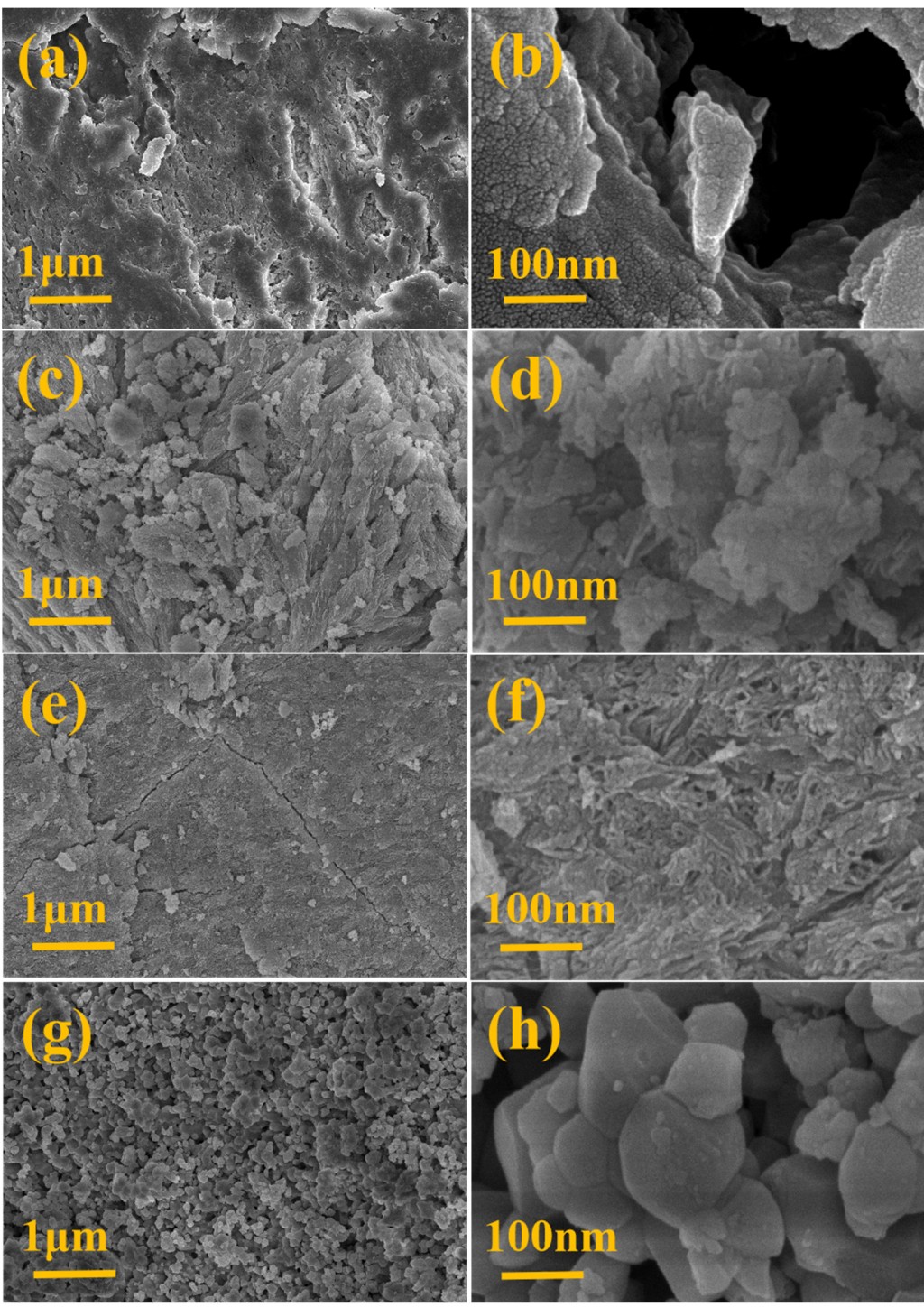

**Figure 1.** The SEM images of BC and BCH samples: RB (**a**,**b**), BCH-300 (**c**,**d**), BC-300 (**e**,**f**), BC-900 (**g**,**h**).

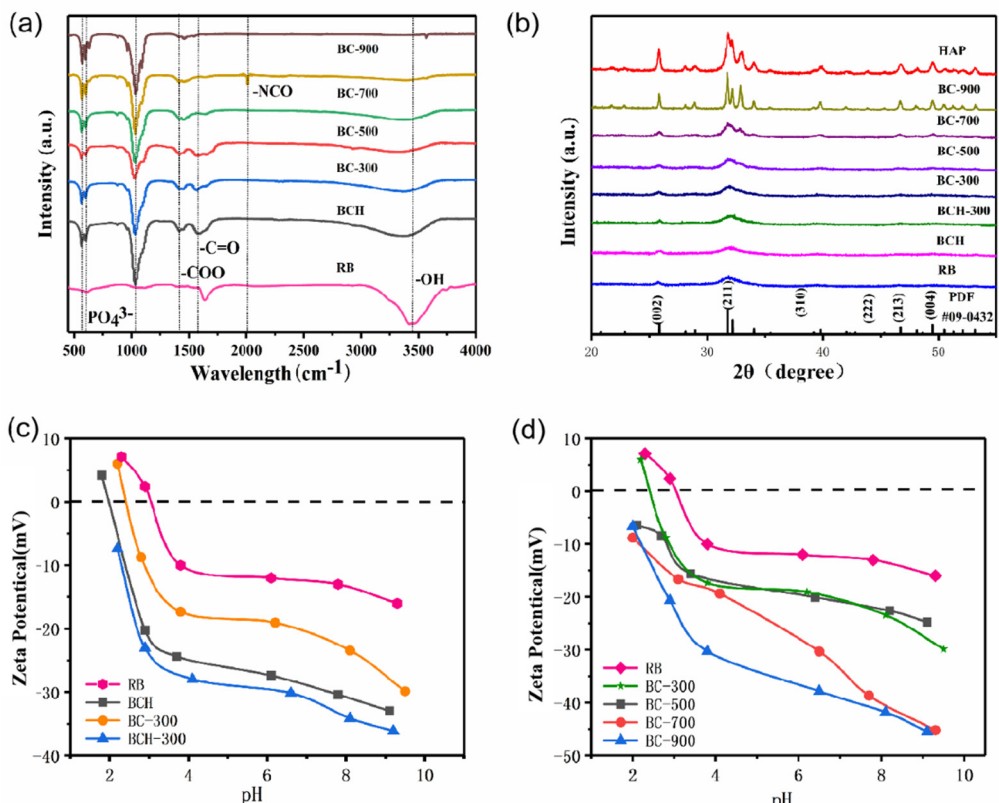

**Figure 2.** (**a**) Fourier-transform infrared spectra, (**b**) XRD patterns, and (**c**,**d**) the zeta potential of BC and BCH samples.

The phase purity and crystal structures of the BC and BCH samples obtained under different pyrolysis temperatures were investigated by XRD. As shown in Figure 2b, the diffraction peaks at 2θ values of 25.93°, 31.82°, 39.56°, 46.64°, 49.53°, and 53.23°, respectively, correspond to the (002), (211), (130), (222), (213), and (004) diffraction peaks of hydroxyapatite [34,35]. The intensity and sharpness of these diffraction peaks increase with the increasing pyrolysis temperature. This phenomenon further indicates that the increasing of the pyrolysis temperature leads to a significantly higher level of crystallinity.

Negative charges on the surface of materials promote electrostatic adsorption, which is conducive to the adsorption of heavy metals. Thus, investigating the surface charge of the pyrolyzed bone char samples is of great importance. According to the literature, surface charges are mainly associated with oxygen-containing functional groups on material surfaces [27]. These oxygen-containing functional groups can release protons, resulting in a negative surface charge.

The zeta potentials (Figure 2c,d) and FT-IR spectra (Figure 2a) of the samples show that with increasing pyrolysis temperature, the surface content of phosphoric acid and other oxygen-containing functional groups increases. This may explain the higher surface negative charge of the samples pyrolyzed at higher temperatures (BC-900). Comparing the unpyrolyzed sample (BCH) and the samples pyrolyzed at low temperatures (BC-300 and BCH-300), it is clear that hydrogen peroxide pretreatment also significantly enhances the surface negative charge. This is also due to the increase in surface phosphate and other oxygen-containing functional groups. Thus, both high-temperature treatment and $H_2O_2$ pretreatment enhance the surface negative charge of these bone char samples.

### 3.2. Adsorption Kinetics

The pseudo-second-order kinetic model was used to fit the sorption data, as presented in Figure 3 and Table S2. The $R^2$ association coefficient is greater than 0.9, which shows that it follows the pseudo-second-order model. Thus, this sorption process is a rate-limiting

process such as chemical adsorption that takes place through valence forces. As a result, the sorption of $Cd^{2+}$ to the active sites of the bone char samples is mainly influenced by chemical reactions such as ion exchange or precipitation instead of physisorption [28], which is in good agreement with the experimental results. For all the bone char sorbents investigated in this work, the quantity of ion-interchanged $Ca^{2+}$ cations gradually increased with increasing sorption time until equilibrium is achieved (Figure S3). A comparison of Figure S3a,b demonstrates that dissolved calcium is closely associated with $Cd^{2+}$ adsorption. It can therefore be speculated that adsorption is partially realized via $Cd^{2+}$ and $Ca^{2+}$ ion interchange [21].

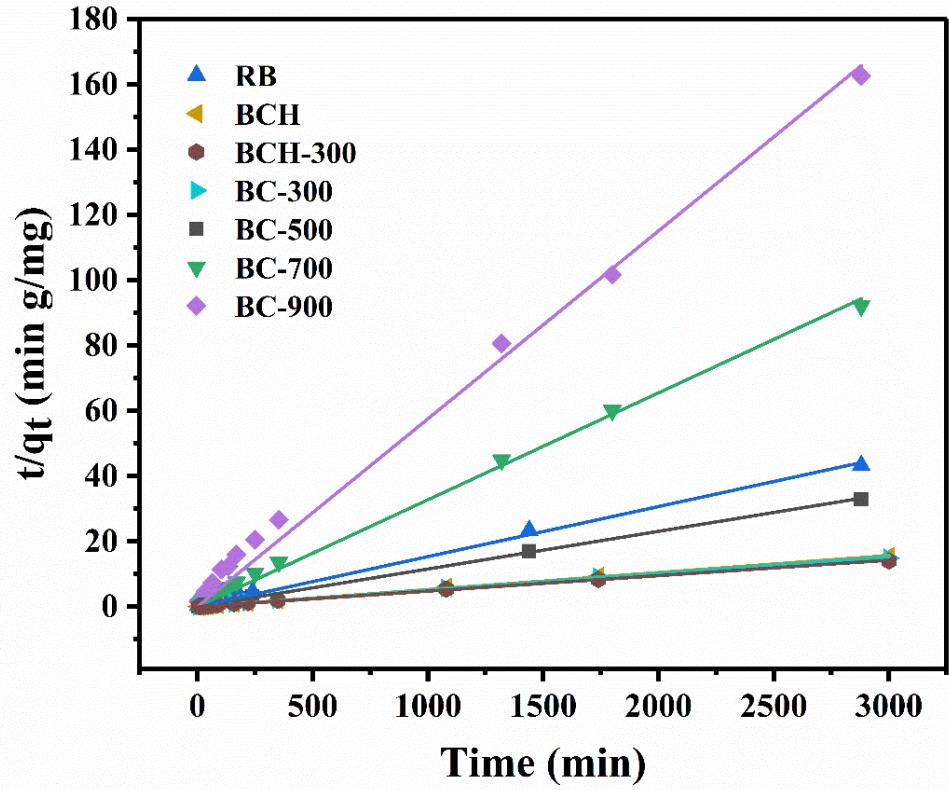

**Figure 3.** The pseudo-second kinetics model on sorption of $Cd^{2+}$ on various BC and BCH samples.

### *3.3. Effect of Solution pH and Ionic Strength*

The pH value is a significant environmental factor that should be investigated to determine the utilization value of materials. This is due to the pH value influencing the surface charge of adsorbents, their adsorption capacity, and the type of heavy metals in an aqueous solution. The pH dependence of $Cd^{2+}$ adsorption on the BC and BCH samples is shown in Figure 4. All the tested sorbents achieved optimum adsorption in the initial solution pH range of 4.0–8.0. The negatively charged surfaces of the bone char samples within this initial pH range electrostatically interact with the positive electric charges of heavy metal groups. The relatively constant absorption of $Cd^{2+}$ is the consequence of a stable final pH, which is dependent on the buffering ability of the adsorbent (Figure 4a). The amount of $Ca^{2+}$ released by the inorganic phase of the bone char sorbents similarly depends on the pH (Figure 4b). The concentrations of $Ca^{2+}$ rapidly decrease with increasing initial pH value from 1.0 to 3.0 and remains constant in the initial pH range of 4.0 to 8.0. The dissolubility of hydroxyapatite, which is enhanced at pH 1.0–3.0, results in high $Ca^{2+}$ ion emission, while $Ca^{2+}$ discharge is mostly caused by the ion interchange with $Cd^{2+}$ ions in the pH range of 4.0–8.0 [20,36].

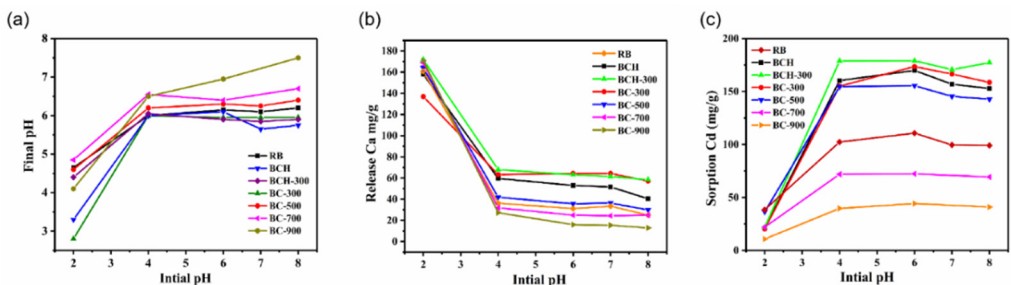

**Figure 4.** Effect of initial pH on the final pH values (**a**), amount of released Ca$^{2+}$ (**b**), and amount of adsorbed Cd$^{2+}$ (**c**).

The presence of various ionic species was used to evaluate the potential mechanisms of Cd$^{2+}$ adsorption on the BC and BCH sorbents. As shown in Figure 5, the amount of Cd$^{2+}$ adsorbed on the BC and BCH samples slightly decrease with the addition of NaNO$_3$ and NaCl. This slight reduction in heavy metal adsorption is potentially due to the high Na$^+$ content in the aqueous solution. The presence of Na$^+$ ions leads to steric hindrance, which impedes Cd$^{2+}$ adsorption. Unlike Na+ cations, Cl$^-$ anions and SO$_4{}^{2-}$ anions have almost no effect on Cd$^{2+}$ adsorption [37], while F$^-$ anions have a very significant promoting effect. This may be associated with the synergistic combination of F$^-$ ions and Ca$^{2+}$ ions. The promoting effect of F$^-$ ions on Ca$^{2+}$ release from the bone char adsorbents follows the order: BCH-300 > BCH > BC-300 > BC-500 > RB > BC-700 > BC-900. According to the XRD results of the materials after adding F$^-$ (Figure S4), the formation of calcium fluoride may promote the ion exchange process of bone char for Cd$^{2+}$.

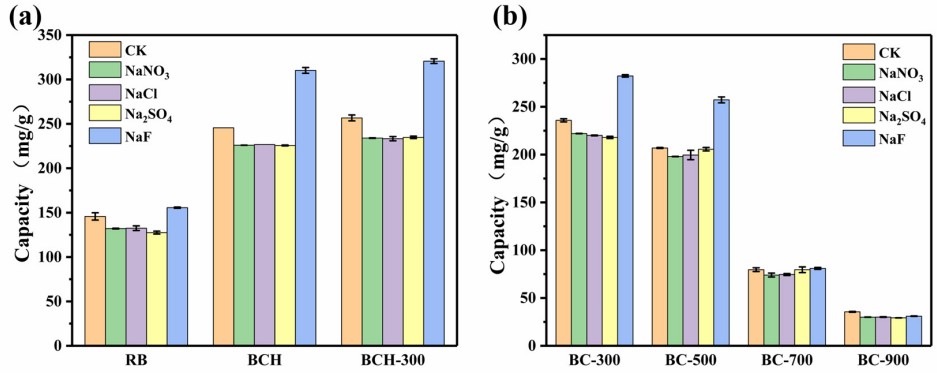

**Figure 5.** Effect of various ionic species on sorption performance (**a**,**b**).

### 3.4. Adsorption Isotherms

Adsorption isotherm was used to investigate the adsorption behavior of Cd$^{2+}$ on the adsorbents. As shown in Figure 6a, sorption isotherms for Cd$^{2+}$ on the BC and BCH samples were obtained at various concentrations. The association coefficients (R$^2$) calculated for the Langmuir isotherm model demonstrate that this model is a closer fit for the adsorption data compared with those calculated for the Freundlich isotherm model. This demonstrates that the Cd$^{2+}$ adsorption sites on the bone char surfaces are homogeneous [34].

The amount of Ca$^{2+}$ discharged from the sorbent increases with the increasing quantity of adsorbed Cd$^{2+}$ (Figure 6b). The molar ratio of Ca$^{2+}$ release and Cd$^{2+}$ adsorption on the same material is quite stable, demonstrating a linear relationship between the discharged cations and adsorbed Cd$^{2+}$. As the cations released from the bone char samples are mainly Ca$^{2+}$, the higher Ca$^{2+}$ concentration proves that ion exchange is an important adsorption mechanism. Moreover, the slope of the linear relationship reflects the proportion of ion exchange relative to the overall adsorption process. The slopes of the different bone char samples are significantly different and follow the order BCH-300 > BCH > BC-300 > BC-

500 > RB > BC-700 > BC-900. This phenomenon indicated that BCH-300 has a better ion exchangeability.

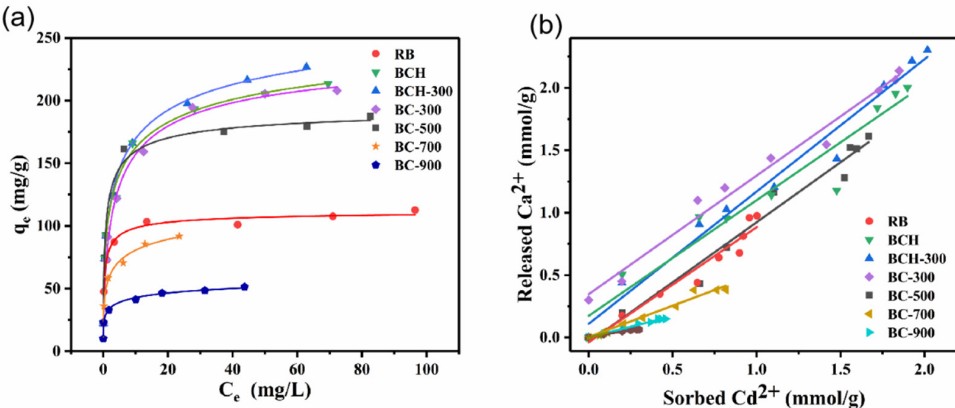

**Figure 6.** The Langmuir model fit of $Cd^{2+}$ adsorption on BC and BCH samples (**a**) and the relationships between $Cd^{2+}$ adsorption and $Ca^{2+}$ release (**b**).

### 3.5. Sorption Mechanisms

To investigate the complex adsorption process of metal ions on the bone chars, FT-IR was carried out on the pristine and Cd-loaded BC and BCH samples. As shown in Figure 7a, the intensities of the FT-IR bands corresponding to the v (–OH) oscillation of hydroxyl groups at 3360 cm$^{-1}$ rapidly increase with $Cd^{2+}$ adsorption. Moreover, the positions of these bands alternate after $Cd^{2+}$ adsorption, indicating the participation of the hydroxyl groups in the $Cd^{2+}$ uptake process [34]. On the other hand, the C=O bands at 1570 cm$^{-1}$ and 1090 cm$^{-1}$ dramatically decrease in intensity after $Cd^{2+}$ adsorption.

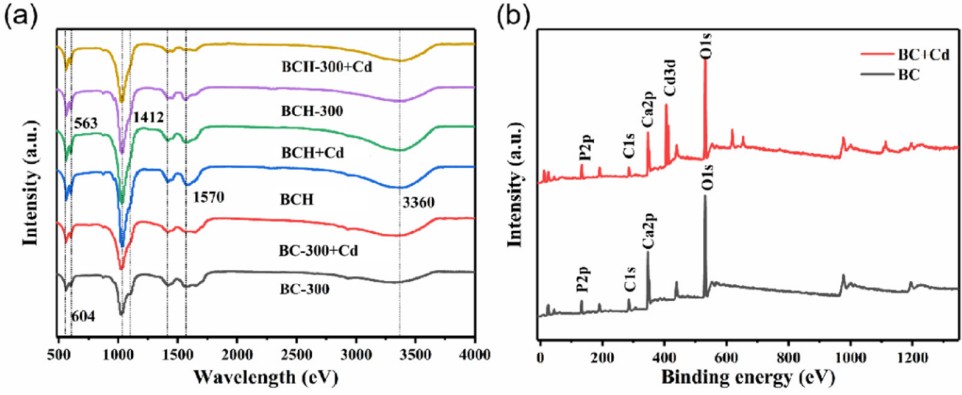

**Figure 7.** Analysis of BC and BCH samples before and after adsorbing Cd: (**a**) FT-IR spectra, (**b**) XPS survey spectra.

The strength of the –COO band at 1453 cm$^{-1}$ and those of the phosphate groups significantly different before and after $Cd^{2+}$ sorption. This band is caused by the interaction between the calcium ions and phosphate groups. Moreover, the Ca–O signal at 565 cm$^{-1}$ also changes due to $Cd^{2+}$ sorption, with an intensity lower than that of the BC spectrum. This result also demonstrates the ion exchange between the cadmium ions in the aqueous solution and the calcium ions originating from the hydroxyapatite of the adsorbent material [36]. In summary, these FT-IR results indicate that the oxygen-containing functional groups of the BC and BCH samples are significantly transformed by $Cd^{2+}$ adsorption, demonstrating the successful chemical combination of cadmium ions with the surface functional groups of the bone char sorbents.

XPS analysis of BCH-300 before and after cadmium sorption was carried out to investigate the chemical transformation on the surface of the bone char samples in more

detail (Figures 7b and S6). Notably, new intense double peaks appear in the survey spectrum of BCH-300 after $Cd^{2+}$ adsorption, indicating the successful adsorption of calcium on the surface of BCH-300 [34,35].

The O 1s spectrum presented in Figure S6 can be deconvoluted into two peaks representing P=O/C=O (531.2 eV) and P–O/C–O (533.0 eV). The intensities of the O 1s spectra of BCH-300 before and after $Cd^{2+}$ adsorption are significantly different, demonstrating the existence of interactions between cadmium and the oxygen-containing functional groups on the BCH-300 surface. Moreover, the peak area ratios of the BC-300, BCH, and BCH-300 O 1s spectra significantly fluctuate after $Cd^{2+}$ adsorption, indicating that these oxygen-containing functional groups play an important role in cadmium adsorption [28,38].

The P 2 p spectra of the BCH-300 specimen present the most significant fluctuation in intensity after $Cd^{2+}$ adsorption. This spectrum broadens after the elimination operation, which demonstrates the influence of the surface phosphate groups on the adsorption process. Figure S6 shows that the carbonate peak significantly changes with $Cd^{2+}$ adsorption as well, indicating that the carbonate binds to cadmium during the sorption process [39].

A comparison of the Ca spectra before and after $Cd^{2+}$ adsorption indicates that the Ca peak significantly declines in intensity, which is potentially due to the loss of Ca–O bonds in the bone char structure as well as changes to the electron density of the Ca atoms that remain after the adsorption process. As previously discussed, this adsorption process is driven by the exchange of calcium ions from the hydroxyapatite component of the bone char with cadmium ions in the solution. This trend may also be related to the interactions between cadmium and phosphate groups [35]. These characterization results confirm that the surface complexation of oxygen-containing functional groups and the exchange of lattice calcium with cadmium ions in solution are important $Cd^{2+}$ adsorption mechanisms [40].

In summary, as shown in Figure 8, the potential mechanisms for $Cd^{2+}$ adsorption are: (a) Ion exchange of lattice calcium with cadmium in aqueous solution; (b) surface complexation between oxygen-containing groups; and (c) electrostatic interactions between the positively charged $Cd^{2+}$ and the negatively charged bone char surface and oxygen-containing functional groups formed after the oxidation of organic matter [38–41].

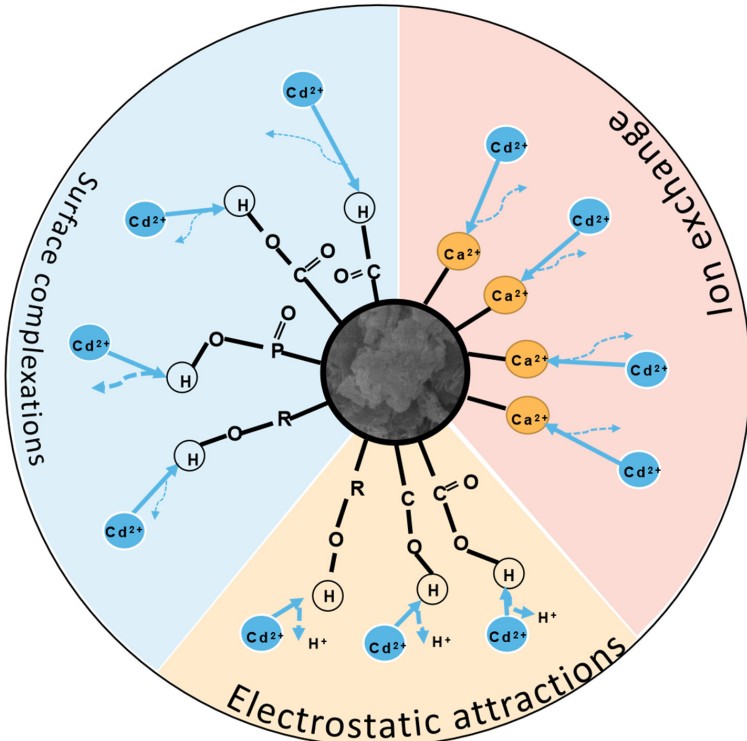

**Figure 8.** Adsorption mechanisms of $Cd^{2+}$ uptake by BC and BCH samples.

### 3.6. Desorption Study

The desorption efficiency of Cd-loaded BC and BCH samples tested with different leaching solutions is shown in Figure 9. It is evident that $Cd^{2+}$ desorption increased with decreasing pH and increasing $Ca^{2+}$ concentration and showed the same trend for all samples.

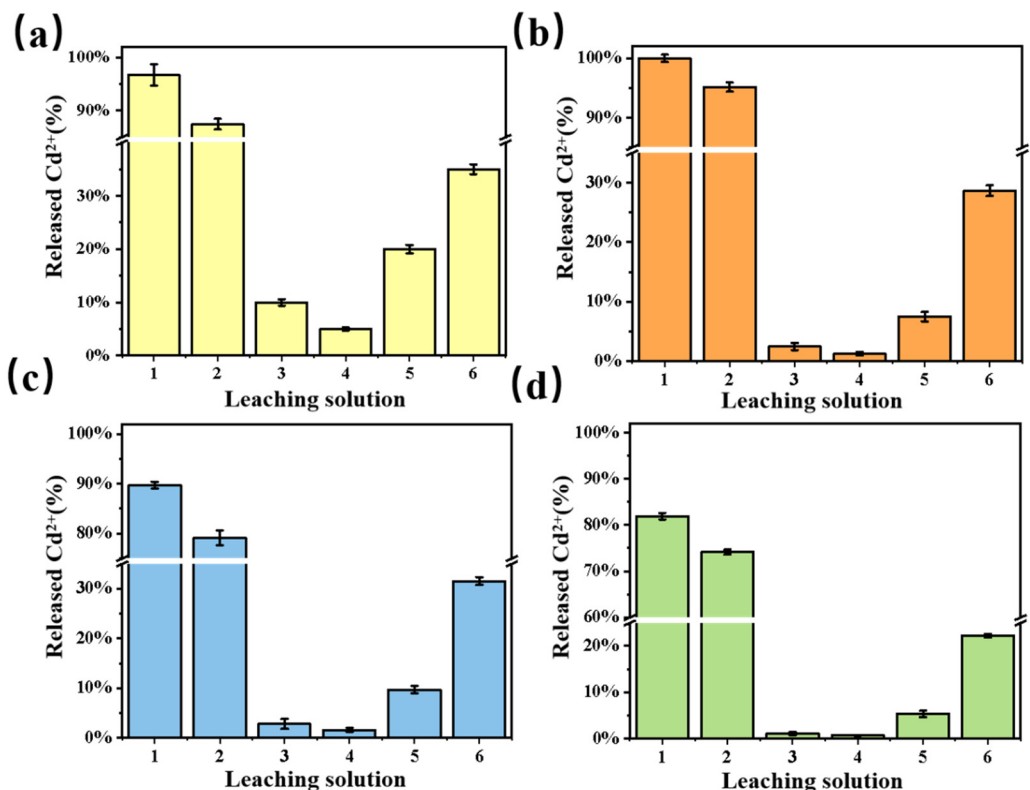

**Figure 9.** Relative percentage of $Cd^{2+}$ ions desorbed from BC and BCH samples: RB (**a**), BC-300 (**b**), BCH (**c**), BCH-300 (**d**). Leaching solutions: (1) pH 1, (2) pH 2, (3) pH 4, (4) pH 6, (5) 25 mmol/L $Ca(NO_3)_2$ and (6) 100 mmol/L $Ca(NO_3)_2$.

The results showed that bone char samples were unstable in a strongly acidic environment (pH 1, 2) and the $Cd^{2+}$ release varied in the range of 74–99%, which was attributed to the solubility of hydroxyapatite in bone char. $Cd^{2+}$ was easily released in the presence of a $Ca^{2+}$ concentration (25 and 100 mmol/L) since the adsorption process occurs mainly through ion exchange. In the common water environment (pH 4, 6), the bone char materials are quite stable. Besides, when comparing the desorption experiments of the samples, it can be seen that BCH-300 is the most stable of all the samples, which can be attributed to the enhancement of surface complexation and electrostatic interactions.

### 3.7. Comparison of BCH with Other Adsorbents

Table 1 showed the results of the adsorption capacity of $Cd^{2+}$ on other materials. Except for the expensive nano-material Nano-Go, the adsorption properties of other materials were significantly lower than that of BCH-300 such as cauliflower leaves biochar, banana peel biochar, oiltea shell, biochar from paper mill sludge, Hickory wood biochar. Moreover, the adsorption capacity of MBC-600 and HAP for cadmium is also obviously weaker than that of the BCH-300, which proves that the new preparation process is easy to synthesize, and has a low cost and high efficiency.

**Table 1.** Compared of cadmium sorption capacity of BCH-300 with other adsorbents.

| Adsorbents Samples | Sorption Capacity (mg/g) | References |
| --- | --- | --- |
| MBC-600 | Cd 165.77 | [28] |
| cauliflower leaves biochar | Cd 73.80 | [42] |
| banana peel biochar | Cd 121.31 | [42] |
| HAP | Cd 49.36 | [43] |
| oiltea shell | Cd 22.40 | [44] |
| biochar from paper mill sludge | Cd 41.6 | [45] |
| graphene oxide (GO) | Cd 530 | [46] |
| Hickory wood biochar | Cd 4.75 | [47] |
| BCH-300 | Cd 228.73 | This work |

## 4. Conclusions

The physicochemical properties and $Cd^{2+}$ adsorption performance of non-pretreated BC and hydrogen peroxide pretreated BCH samples prepared under varying pyrolysis temperatures were thoroughly investigated. Due to their excellent buffering properties, these bone char sorbents showed good $Cd^{2+}$ adsorption performance in solutions with an initial pH of 4.0–8.0. The adsorption performance of the bone char sorbents slightly decreased in the presence of $Na^+$ ions, although the $Cl^-$ and $SO_4^{2-}$ ions did not significantly affect $Cd^{2+}$ adsorption. However, $Cd^{2+}$ adsorption was significantly enhanced in the presence of F ions, which was potentially related to the promotion of $Ca^{2+}$ and $Cd^{2+}$ ion exchange by the $F^-$. The adsorption kinetics of the $Cd^{2+}$ adsorption process followed a pseudo-second-order model, and the adsorption isotherms followed a Langmuir isotherm. The main mechanisms of $Cd^{2+}$ adsorption included the ion exchange of lattice $Cd^{2+}$ from the bone char with cadmium ions from the aqueous solution, surface complexation between oxygen-containing groups, and electrostatic interactions. Thus, hydrogen peroxide pretreatment combined with low-temperature pyrolysis was successfully employed to reduce organic matter content while not excessively increasing crystallinity. BCH-300 exhibited the most significantly enhanced adsorption capacity and the highest adsorption efficiency for $Cd^{2+}$ compared with the other tested bone char samples. This was due to its low crystallinity, low surface zeta potential, and large specific surface area. These results demonstrated that the bone char, which was prepared by a low-cost and environmentally friendly hydrogen peroxide pretreatment combined with low-temperature pyrolysis showed great potential for the remediation of Cd-containing aquatic environments.

**Supplementary Materials:** The following are available online at https://www.mdpi.com/article/10.3390/pr10040618/s1, Text S1: Characterization of BCs and BCHs. Text S2: Batch sorption experiments. Text S3: Analytical methods. Figure S1: The SEM images of BC and BCH samples: BCH(a, b), BC-500(c, d), BC-700(e, f). Figure S2: DTA/TGA analysis of bone sample. Figure S3: Effect of contact time on amounts of $Cd^{2+}$ adsorbed (a) and $Ca^{2+}$ released from adsorbents (b). Figure S4 The XRD spectra of adsorbent after adsorbed $Cd^{2+}$ with 10mmol/L NaF. Figure S5: The relationships between equilibrium $Cd^{2+}$ concentrations and the amounts of $Ca^{2+}$ released from adsorbents. Figure S6: The spectra of BCs before and after adsorbed Cd: XPS spectra: (a) P2p spectra, (b) O1 s spectra, (c) Ca 2p spectra,(d) C 1s spectra. Table S1: Properties and adsorption capacity of bone char samples. Table S2: Pseudo-first-order and pseudo-second-order kinetic parameters for $Cd^{2+}$ adsorbed on adsorbents form aqueous solution. Table S3: Langmuir and Freundlich parameters for $Cd^{2+}$ adsorbed on adsorbents from aqueous solution.

**Author Contributions:** Q.G.: Conceptualization, Methodology, Formal analysis, Investigation, Writing—original draft, Visualization. H.T.: Formal analysis, Investigation. L.J.: Writing—review and editing. M.C.: Formal analysis, Investigation. N.Z.: Resources. P.W.: Resources, Supervision, Project administration, Funding acquisition. All authors have read and agreed to the published version of the manuscript.

**Funding:** The authors appreciate financial support from the National Natural Science Foundation of China (Grant Nos. 41972037, 41673092).

**Institutional Review Board Statement:** Not applicable.

**Informed Consent Statement:** Not applicable.

**Data Availability Statement:** Not applicable.

**Acknowledgments:** The authors would like to thank the editors and anonymous reviewers for their valuable and constructive comments related to this manuscript.

**Conflicts of Interest:** The authors declare no conflict of interest.

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
