# Peer review of "Sorption of Cd2+ on Bone Chars with or without Hydrogen Peroxide Treatment under Various Pyrolysis Temperatures: Comparison of Mechanisms and Performance"

_processes, doi:10.3390/pr10040618_

Round 1

Reviewer 1 Report

In this manuscript, the authors study the Sorption of Cd2+ on bone chars with or without hydrogen peroxide treatment under various pyrolysis temperatures: Comparison of mechanisms and performance. This work sounds very interesting and meaningful, and the analysis is reasonably clear, but there are still a few concerns regarding this article, therefore, I suggest it for publication in this journal after the following major revisions:

  1. It can be compared with the existing research to highlight the research advantages of this paper.
  2. What is the long-term adsorption performance of the adsorbent in this experiment.
  3. Some grammatical errors are found. A proof-reading of the article will enhance the quality of the paper.
  4. The format of images and notes should be consistent.

Reviewer 2 Report

Dear editor,

the following issue have to be corrected in order to be published:

  • graphical abstract should be corrected and improved
  • put also the comparasion values in abstract
  • use subscript and superscript in all manuscript
  • why did you use inert atmosphere ?
  • why use only for 300oC the H2O2?
  • raw 100, why did you wash them again after pyrolysis?
  • why did you heat treated the samples if BCH has good adsorption also without heat treating ?
  • raw 135:   DOC and EDS  please add full name
  • why do you add Ca2+ cations aslo in manuscript? please change title then and complete the results
  • XPS analysis of BCH-300 or BC-300 ? in figure you write BC and in text BCH
  • please check for typos in all manuscript there are a lot of mistakes.
  •  

Round 2

Reviewer 1 Report

The authors took into consideration the reviewers’ comments, they analyzed simulation result and answered to the questions. They addressed most of reviewers’ comments and took them into account by modifying the manuscript and the supplementary information. They nicely answered the questions by providing data, analysis and discussions. The manuscript has been changed and is now better and clearer. Overall I think this work can now be published in Processes. This is why I recommend accepting this manuscript.